# Non-Targeted Screening Approaches for Profiling of Volatile Organic Compounds Based on Gas Chromatography-Ion Mobility Spectroscopy (GC-IMS) and Machine Learning

**DOI:** 10.3390/molecules26185457

**Published:** 2021-09-08

**Authors:** Charlotte Capitain, Philipp Weller

**Affiliations:** Institute for Instrumental Analytics and Bioanalytics, Mannheim University of Applied Sciences, 68163 Mannheim, Germany; c.capitain@hs-mannheim.de

**Keywords:** gas chromatography ion mobility spectroscopy (GC-IMS), volatile organic compounds (VOCs), non-targeted screening (NTS) using machine learning

## Abstract

Due to its high sensitivity and resolving power, gas chromatography-ion mobility spectrometry (GC-IMS) is a powerful technique for the separation and sensitive detection of volatile organic compounds. It is a robust and easy-to-handle technique, which has recently gained attention for non-targeted screening (NTS) approaches. In this article, the general working principles of GC-IMS are presented. Next, the workflow for NTS using GC-IMS is described, including data acquisition, data processing and model building, model interpretation and complementary data analysis. A detailed overview of recent studies for NTS using GC-IMS is included, including several examples which have demonstrated GC-IMS to be an effective technique for various classification and quantification tasks. Lastly, a comparison of targeted and non-targeted strategies using GC-IMS are provided, highlighting the potential of GC-IMS in combination with NTS.

## 1. Introduction

Quality control and early detection of hazard chemicals, allergens, or biological contaminants are critical to ensure product safety. Environmental pollutants, pesticides, or toxins, among others, can compromise food safety and pose a public health risk [1]. Furthermore, food adulteration and food fraud, accelerated by globalization, continue to cause economic losses and customer dissatisfaction and emphasize the need for robust, inexpensive, and fast analytical methods [2]. While new scientific findings continuously identify potential hazardous or allergenic compounds [3], commonly employed methods, which focus on the detection and identification of a particular compound or class of compounds, lack the ability to identify new or unknown compounds. Due to the inherent diversity of biogenic samples, as observed in food analysis, and the chemical complexity of the sample matrices, analysis often requires advanced sample preparation strategies [4]. For systematic monitoring of product quality, it is therefore desirable to develop analytical methods capable of discovering unknown or non-targeted compounds from the complex sample matrices. This approach, also referred to as NTS, requires comprehensive extraction and analysis of compounds of interest. Analysis of the volatile organic compounds (VOCs) of samples, also known as VOC profiling, allows for the detection of compounds in complex sample matrices without the need for detailed a priori knowledge of the molecular composition. Due to its high sensitivity and resolving power on the one hand and its simplicity and robustness on the other, ion mobility spectrometry (IMS) has gained popularity for the analysis of VOCs [5]. Moreover, gas chromatography coupled to ion mobility spectroscopy (GC-IMS) has been shown to be an easy-to-handle and yet highly effective tool for VOC profiling [6]. As a result, non-targeted VOC profiling based on GC-IMS in combination with machine learning has emerged as a promising method for sample monitoring.

Since the 1970s, when IMS was first known as ‘plasma chromatography’, IMS has developed into a highly sensitive technique for the analysis of VOCs at ultratrace concentration levels, which accounts for additional information regarding the ion’s mobility [7,8,9]. Due to the robust and easy-to-handle instrumentation, a wide range of application fields have been found for IMS today, such as food flavor analysis [5], process monitoring [10,11], and quality control [12], as well as detection and quantification of warfare agents [13] and explosives [14,15].

With IMS, analytes are first ionized in the ionization region of the instrument. The most common ionization method is the atmospheric pressure chemical ionization [16] by beta emitters, which frequently use nickel-63 (Ni-63) [11,15,17,18] or the less hazardous beta-emitting tritium (H-3) [19] or alpha-emitting americium-241 (Am-241) [20,21]. Other ionization methods are atmospheric pressure photo ionization (APPI) [22], which uses ultraviolet light (UV) [23,24] or corona discharge (CD) atmospheric pressure chemical ionization [17,25,26,27], where a high electric field between a needle and a metal plate or discharge electrode is used. Yet another method is the laser desorption/ionization technique (LDI), which employs a laser pulse as ion source [28].

According to the European Union directive, the exemption limit for the total activity for tritium was set to 1 GBq [29]. Therefore, the usage of a low-radiation tritium ion source with an activity of 300 MBq is not subject to authorization, hence leading to a broad adoption of tritium ion sources in a number of commercially available systems on the market [30,31,32,33,34]. Beta particles, which are emitted by the tritium source, initiate a gas-phase reaction cascade of the drift gas (nitrogen or air), resulting in predominant proton-water clusters H^+^[H_2_O]_n_, which are commonly referred to as ‘reactant ions’ [35]. The number of water molecules (n) depends on the gas temperature and the moisture content of the gas atmosphere [8]. Depending on the proton affinity, molecules entering the ionization region react with the reactant ions to protonated monomers MH^+^[H_2_O]_n−x_, while decreasing the intensity of the reactant ion peak (RIP). At higher analyte concentrations, proton-bound dimers M_2_H^+^[H_2_O]_m−x_ are formed by the attachment of additional analyte molecules. When the concentration further increases, the formation of higher molecular cluster ions, such as trimers or tetramers, is possible; however, due to their low stability and short lifetime, higher molecular cluster ions are rarely observed [36]. In general, nonlinear behaviors are observed for the ratio of the RIP and the distribution between the protonated monomer and the proton-bound dimer [36,37]. The principles of a drift-time IMS including a H-3 ionization source are shown in Figure 1.

Subsequent to ionization, the analyte ions enter the drift region, where they are accelerated towards the detector, typically a Faraday plate, and are separated by their drift time (or mobility) in an electrical field at ambient pressure. The ions are slowed down by the collision with counterflowing drift gas molecules in the collision cross-section (CCS). Due to an equilibrium between the acceleration by an electric field and deceleration by the collision with the drift gas molecules, the ions move with a constant velocity to the detector. Depending on the characteristic mass, charge, and structure, the ions are separated in the drift tube and reach the detector at different drift times [39]. For identification of the analyte, the inverse of the measured drift time is normalized to the drift length and the electric field resulting in the spectrum of ion mobility. The reduced ion mobility K_0_ (see Equation (1)), which is independent of ambient conditions and experimental setup, is obtained after further normalization to pressure and temperature.
(1)$$K_{0}=\frac{L}{E\cdot t_{D}}\cdot \frac{p}{p_{0}}\cdot \frac{T_{0}}{T}$$

With

K_0_ = reduced ion mobility in cm²V^−1^s^−1^L = drift length in cmE = electric field strength in Vcm^−1^t_D_ = drift time in sp = pressure of the drift gas in hPap_0_ = ambient pressure: p_0_ = 1013.2 hPaT = temperature of the drift gas in KT_0_-ambient temperature: T_0_ = 273.2 K

Instead of measuring temperature and pressure, the normalization is often carried out using the known mobility of the ions produced in the pure drift gas or by adding a reference analyte [40]. The signal intensity is proportional to the concentration and enables the quantification in ppb_v_ (for some compounds even ppt_v_) levels within a few milliseconds.

The state-of-the-art IMS technologies can be classified into time-dispersive, space-dispersive, and trapping technologies [41]. Time-dispersive IMS separates ions as a function of their mobility in a neutral gas, whereas space-dispersive IMS separates ions by the ratio of low-field to high-field mobilities [42]. Examples of time-dispersive IMS are drift tube ion mobility spectrometry (DTIMS) and travelling tube ion mobility spectrometry (TWIMS). High-field asymmetric waveform ion mobility spectrometry (FAIMS) and differential ion mobility spectrometry (DIMS or DMS) are examples for space-dispersive techniques [43]. The third class is represented by trapped ion mobility spectrometry (TIMS), which contains a trapping technology able to confine and release ions.

IMS alone has been applied for quantification [11,15,17,27] and classification [5] tasks in controlled environments. However, due to the inherent diversity of biogenic samples, the applications of IMS with direct sample introduction are often not sufficient, requiring prior purification or separation. The commonly used purification methods for VOC profiling in combination with IMS are solvent extraction [20,26,44] and solid-phase microextraction (SPME) [13]. SPME devices are constructed of a silica fiber coated with a thin layer of a suitable polymeric sorbent or immobilized liquid, used for the direct extraction of analytes from gaseous and liquid media [45]. While SPME coupled to IMS has been successfully used for quantification tasks, such as the detection and quantification of precursors and degradation products of chemical warfare agents [13], SPME is commonly extended by column separation techniques [18,46,47].

To avoid clustering in the ionization or drift region, IMS devices are commonly coupled to column separation techniques, such as liquid chromatography (LC) or gas chromatography (GC). Column separation coupled to drift-time IMS separates analytes into two orthogonal features, first the retention time through chromatography and second, the drift time or mobility through IMS, resulting in a two-dimensional (2D) highly resolved GC-IMS spectrum [6,38]. In LC analysis, any soluble compound can be separated, but sample preparation is a critical step for the data quality [48]. A comprehensive extraction method which enables the extraction of a wide range of compounds with minimized potentially interfering coextractives is needed for NTS approaches, since unspecific compounds are being targeted [49]. LC-MS in combination with NTS has been applied for the detection of food contaminants and environmental hazards [50,51].

In GC analysis, the volatility of a sample is a prerequisite. Headspace (HS)-based techniques allow for the analysis of untreated samples, avoiding the time-consuming sample pretreatment steps [52]. The analysis of non-volatile samples may be achieved through the derivatization with a functional group onto the molecule of interest. Although the modification of the functional group enables the analysis of compounds that otherwise could not be easily monitored by GC, NTS approaches usually do not incorporate derivatization, in particular due to the high level of variance.

The advantages of GC-IMS in comparison to established techniques, such as mass spectrometry, are its simple and inexpensive design primarily due to being operated at atmospheric pressure and hence not requiring vacuum pumps [8]. Furthermore, the use of radioactive ionization sources allows for portability, miniaturization, and mechanical robustness and therefore is suitable for field and benchtop applications [52]. Due to efficient ionization, in combination with its fast and sensitive detection, IMS is a universal technique for the analysis of organic and inorganic molecules, atoms, or particles [38]. One potential challenge of IMS analysis is that spectra may contain interference due to widespread ionization, which results in low selectivity. The addition of suitable dopant substances, however, has been shown to overcome these limitations [53,54]. A nonlinear concentration range was previously described for IMS, requiring the careful monitoring of sample concentration to avoid sample saturation. Furthermore, the separation, which is based on CCS, often provides limited information regarding specific qualities concerning size and shape of analytes. However, the drawback of interference caused by spectral complexity and nonlinearities can be overcome by using computer-based analysis tools [55].

The complexity of biological samples results from the presence of a variety of compounds, which provide in their entirety a characteristic HS-GC-IMS spectrum, often referred to as the VOC profile or ‘fingerprint’ [56,57]. HS-GC-IMS has been demonstrated to be an effective technique for the evaluation of VOC profiles of biological samples due to its simple system setup, robustness, and price [44,58,59,60,61]. The chemical fingerprinting of food and beverages in combination with chemometric analysis is widely used for food authentication and ultimately to identify food adulteration and fraud [62]. Furthermore, the VOC profile is influenced by production processes as well as storage conditions. Consequently, process control and quality assurance, such as the control of food freshness or food safety, are topics of interest for NTS using HS-GC-IMS [63,64] techniques.

## 2. Motivation for Non-Targeted Screening Using HS-GC-IMS

Labelling fraud, e.g., of organic certifications or geographic origin, is the most common type of fraud in agricultural and food markets [65,66]. According to the European Commission, honey and olive oil are particularly affected by mislabeled botanical origin, as well as dilution with inferior or less expensive products [2,67]. Moreover, food adulteration and food fraud have led to cases of economic loss and may pose health risks [2]. The detection of food fraud or adulteration often involves the identification of compounds of unknown molecular composition. Since no identified chemical markers or sets of markers are commonly accessible for a target-based analysis, an analytical approach covering a multitude of parameters in parallel paired with strong discrimination power is required. The currently used methods to determine quality and authenticity, such as sensory analysis and physicochemical analysis [68], are time- and resource-consuming, while lacking sensitivity as well as prediction accuracy, not at least due to univariate analysis. To overcome the limitations of traditional, wet-chemistry-based assays, targeted and non-targeted approaches using chromatographic methods [69,70], often in combination with mass spectrometry [71,72,73], as well as infrared (IR)-based spectroscopy [74,75], and proton nuclear magnetic resonance (^1^H NMR) spectroscopy [76,77] have been discussed for various applications. However, to obtain the required reproducibility needed for chemometric analysis, time-consuming sample preparation, including precise adjustments of pH, water content or particle size, have been reported in combination with the mentioned methods. Furthermore, the high costs of ownership and maintenance, as well as the requirement for expert knowledge, may limit applications. Finally, high-end instrumentation also requires suitable laboratory infrastructures, which are usually not available at the point of care. Thus, robust, inexpensive, and fast analytical methods, such as HS-GC-MS, are needed, which require little or no sample preparation but deliver high selectivity.

Application examples for HS-GC-IMS with NTS:

A plethora of studies have shown the potential of HS-GC-IMS in combination with NTS for monitoring food quality or confirmation of geographical or botanical origin, despite the complexity of the samples. For example, HS-GC-IMS with NTS has been widely applied for the classification of olive oil between high-priced type 1 extra-virgin olive oil (EVOO), medium-priced type 2 virgin olive oil (VOO or OO), and non-edible type 3 olive oil, also known as pomace olive oil (POO) or lampante (virgin) olive oil (L(V)OO) [32,33,78,79]. Furthermore, HS-GC-IMS with NTS was successfully used for reliable classification of geographical origins for both olive oil (EVOO) [34,80,81] and wine [30]. Moreover, HS-GC-IMS with NTS was applied for the classification of honey according to botanical origin [52,81,82], as well as for the detection and quantification of honey adulterated with sugar cane or corn syrups [83,84]. Recently, HS-GC-IMS with NTS has been applied to assess the freshness of food [85] and for the detection of mold formation on milled rice [86], peanut kernels [87], and wheat kernels [88]. Further examples of recent studies using HS-GC-IMS with NTS are provided in Table 1.

## 3. NTS-Workflow

NTS aims to identify the compounds of unknown molecular composition. The workflow for NTS generally consists of sample preparation, instrumental analysis, and post-acquisition data processing [93]. Since little or no a priori knowledge of the chemical structures and behavior of compounds is required, NTS approaches benefit from gentile sample preparation, robust instrumental analysis, and standardized data processing. The workflow for NTS using GC-IMS is shown in Figure 2. The first step, data acquisition, involves sample preparation and subsequent extraction and separation of VOCs. The collected data are then preprocessed and analyzed in the data-processing step. Since no pre-existing knowledge is used, the entire spectral fingerprint obtained by HS-GC-IMS analysis is subject to data analysis and classification or quantification models being built using machine learning tools. In the third step, model interpretation, key compounds are identified, which are extracted through back projection of loadings. Complementary analyses, such as sensory analysis of GC-MS measurements, provide further insight into sample composition and can be used to improve model accuracy. The model coherency is evaluated and finally applied for benchtop profiling.

### 3.1. Data Acquisition

In untargeted approaches, sample preparation strategies need to be suitable for a variety of matrices and enable the extraction of a wide range of compounds [49]. In order to minimize the extraction-related formation of artifacts, static headspace extraction (SHS) is commonly used for the analysis of volatile compounds in samples of different origin [94]. The limited sensitivity and the bias related to the extraction of low-volatile compounds are generally considered as the main limitations of SHS methods, which can be overcome through dynamic headspace extraction (DHS) [95]. If the preconcentration of analytes is needed for analysis, high-concentration-capacity HS techniques, such as SPME can be used, where the selective isolation of compounds of interest from samples with minimal matrix contamination is crucial [95,96,97]. However, SPME is less commonly applied to IMS with NTS, possibly due to drawbacks which have been associated with commercially available extraction phases, such as the low recovery of polar compounds, insufficient matrix-compatibility, and the reduced life-time and robustness of the coating against matrix macromolecules [98].

VOC profiling based on IMS in combination with NTS has been shown to be an effective tool for various classification tasks, such as the differentiation between various green teas with chestnut-like flavor [5]. Using PLS-DA, the authors were able to obtain an accuracy of 95.6% for the classification of tender, pure, and roasted green tea aromas. However, due to the complexity of many biogenic samples, IMS with direct sample introduction is often not sufficient for NTS approaches, requiring extraction and/or column separation techniques. Shuai and coworkers analyzed adulterated flaxseed oils after n-hexane extraction using a handheld IMS and subsequent data analysis. A recursive support vector machine (R-SVM) led to a model with an accuracy of 93.1% for the identification of adulterated flaxseed oil [44].

To address the complexity of biogenic samples, pre-IMS separation techniques are commonly applied. To avoid the clustering and formation of heteromers in the ionization or drift region, IMS devices are commonly coupled to GC. The spectra obtained by HS-GC-IMS are 2D, with the GC retention time as the first dimension and the IMS drift time as the second dimension. The complexity of biological samples results from a plethora of compounds, which provide in their entirety a characteristic GC-IMS spectrum that is often referred to as VOC profile or ‘fingerprint’ [56,57]. For GC separation in combination with IMS and NTS, both the capillary column (CC) and multicapillary column (MCC) have been coupled to IMS. Among others, nonpolar CCs, such as fused silica SE-54-CB (94% methyl, 5% phenyl, 1% vinyl silicone) [32,33,86,89,92], as well as polar CCs, such as DB-225 (25% phenyl, 25% cyanopropyl methyl siloxane) [79,81], were used for HS-GC-IMS with NTS. For MCCs, which are composed of a large number of parallel glass capillaries (~1000), nonpolar OV-5 (5% diphenyl, 95% dimethylpolysiloxane) was used [78,99]. Compared to the ordinary CC, MCCs allow the separation to be carried out at an elevated speed and can be operated at increased carrier gas flows, which may pose an advantage for IMS analysis [100]. However, Garrido-Delgado and coworkers obtained the better predictive accuracy for the classification (*k*NN classifier with *k* = 3) between different qualities of olive oil (EVOO, OO, and LVOO), when CC-IMS (83%) versus MCC-IMS (79%) was employed for the separation of the VOCs [32].

The data acquisition is finalized with the evaluation of the suitability of the collected data for factorial analysis. This sampling adequacy is commonly determined using the Kaiser–Meyer–Olkin (KMO) test, which is a measure of the proportion of variance among variables [101].

### 3.2. Data Processing and Model Building

The ion velocity and thus the signal position are highly dependent on temperature and ambient pressure; therefore, the drift-time alignment is a crucial preprocessing step [36,102]. For baseline correction, drift times are often normalized to the RIP position, which is, however, not sufficient for the determination of absolute mobilities [103]. Alternatively, a reference peak/substance can be used for alignment, which is recommended when GC-IMS measurements are performed with long separation columns. In a second preprocessing step, multiple measurements are averaged, and the background noise is subtracted. Scaling methods, such as unit variance (UnVa) scaling, Pareto (Par) scaling, and mean centering (Ctr) scaling, can be alternatively used for data set normalization [5]. In UnVa scaling, the variance of each variable is unified using the standard deviation, while in Par scaling, the square root of the standard deviation is used for normalization, whereas Ctr scaling emphasizes the analysis of data fluctuations and the large-fold change in the data [82]. For further noise reduction, smoothing algorithms, such as Savitzky-Golay or Gaussian smoothing, are applied to the spectra [52]. Interfering peaks and nonrelevant areas can be removed by the careful selection of relevant GC retention time and drift-time ranges. Prior to data analysis, the 2D spectra (GC retention time × drift time) are unfolded into arrays, which are then concatenated to the final data set (samples × measurements). For supervised data analyses, the data set is finally split into training and test sets, where the training set is used for model building and the test set for model validation.

Nonlinear behavior has been described for the ratio of the RIP and the distribution between the protonated monomer and the proton-bound dimer [36,37,104]. By increasing the molecular concentration of the analyte, the monomer peak decreases, while the dimer peak increases. These typical nonlinear monomer-dimer distributions in IMS are often accompanied by tailing effects of the monomer or dimer peak [105], requiring careful consideration for quantification. In univariate regression (UR), the peak area or intensity of a single peak is correlated against concentration [46]. Due to the nonlinear behavior between the monomer and dimer peak, linearity can only be approximated for narrow concentration ranges when using the regression analysis based on single-peak analysis [36]. Occasionally, the sum of the volume of monomer and dimer is correlated against the concentration [106]; however, in complex mixtures, quantification is further complicated by overlapping peaks, competitive ionization situations between coeluting analytes and the occurrence of multicomponent cluster ions (heterodimeric ions), which are composed of coeluting analytes [37].

#### 3.2.1. Exploratory Data Analysis and Machine Learning Techniques

Due to the complexity of food matrices, abstract terms such as ‘food quality’ and ‘authenticity’ are the sum of multiple characteristics, which implies that the correlation to a single analyte or analytical technique is often not sufficient or even impossible; hence, multivariate variate data analysis (MVA) techniques are required [81]. MVA approaches can be divided into exploratory, classification, and quantitative regression methods. Exploratory methods, such as PCA or HCA, are unsupervised and typically used for pattern recognition, whereas classification methods such as PCA-LDA, *k*NN, or PLS-DA are supervised methods. In this context, PLS-DA is a special case of classification, as it basically uses a regression approach with class boundaries instead of single values, as in quantitative regressions. For sample quantification, the latter methods, such as PLS regression, Kernel-PLSR, MCR-ALS, or ANN are commonly applied.

Principal component analysis (PCA) is a powerful technique for unsupervised discovery of patterns in data, which is further used for dimension reduction [107]. The information extracted from a data matrix is explained by principal components (PCs), which are orthogonal (mathematically independent) to each other. Since PCA models are predicted without labels or validation by test data, they are generally considered unsupervised. Unsupervised statistical methods are exploratory methods that can be used to study data structures and search for clusters of samples [108]. Hierarchical cluster analysis (HCA) of PCA models in a tree-like diagram (dendrogram) is, e.g., used for the visualization of multivariate association and sample similarities [109]. An extension of PCA for processing three-dimensional (3D) data is provided by multiway principal component analysis (MPCA) [110], which has been applied for the feature extraction of GC-IMS matrices, without prior transformation of the 2D data [99]. Further alternatives to PCA and PLS are models based on PARAFAC [111] or Tucker3 [112], which may also be used with 3D data.

Compared to unsupervised techniques, which provide predictions without labels or target variables, supervised techniques aim to build models able to predict target variables. In supervised learning, several data points or samples are described using predictor variables or features and target variables. For classification tasks, the scores obtained by the unsupervised exploratory analysis are combined with subsequent supervised pattern recognition techniques to distinguish samples according to defined categories. Among PCA-based qualitative methods are linear discriminant analysis (LDA) and k-nearest neighbors (*k*NN). Whereas PCA-LDA maximizes the interclass variance, *k*NN assigns the category most common among the *k*-nearest neighbors. The downside of using PCA-based methods is that the correlation between dependent and independent variables are not considered during PCA analysis, which can result in the loss of information included in higher PCs [107]. An alternative is provided by partial least squares (PLS), where the scores are calculated by considering the relationship between the independent and dependent variables.

Other supervised methods used in NTS with HS-GC-IMS are gradient boosting (e.g., XGBoost) [31], decision tree classification (Tree) [91], logistic regression (Regressor) [91], orthogonal partial least-squares discriminant analysis (OPLS-DA), quadratic discriminant analysis (QDA) [30], or soft independent modeling of class analogy (SIMCA) [82]. Furthermore, nonlinear classifications are often performed using support vector machines (SVMs). By using the kernel trick, which transforms the input data into high-dimensional feature spaces, SVMs can perform nonlinear classifications, in addition to performing linear classification [113]. This is of particular importance when no linear hyperplanes are separating the respective classes.

#### 3.2.2. Model Performance and Validation

The performance of a model is usually measured as ‘accuracy’, which is the fraction of correctly classified samples. The classification accuracy determines the fraction of correctly classified samples for a given sample set. The classification accuracy, however, is susceptible to overfitting and thus should only be used as reference. To prevent overfitting, the data set is split into training and test data. The ratio between training and test data, which is commonly referred to as ‘train–test split’, is usually between 2:1 [88] and 4:1 [33], and sometimes as low as 6:1 [31]. The test set is used to determine the prediction accuracy, which is usually lower than the classification accuracy but more meaningful [88]. For small and inhomogeneous data sets, a single split of the data into training and test set may give misleading results [114]. An alternative model validation is therefore provided by resample methods, such as cross validation (CV), bootstrapping, or permutation testing, where multiple random subsets are generated. For the CV of samples, a subset of the data is held out for use as a validation set, and a model is fit to the remaining data (training set) and used to predict the validation set. The process of generating a subset of data, model fitting, and evaluation is performed repeatedly, and an overall prediction accuracy is determined by averaging the quality of the predictions across the validation sets. Leave-one-out CV leaves out a single observation at a time, while k-fold CV splits the data into k subsets, which are one by one held out as the validation set [107]. For NTS using HS-GC-IMS, 10-fold CV is commonly performed [31,52], alongside with leave-one-out CV [80,85]. Bootstrapping is a resampling method that can be used as an alternative to CV to estimate the prediction performance of a model with a low number of training samples. Due to drawing with replacement, a bootstrapped data set contains the same number of cases as the original data set, and it can contain multiple instances of the same original cases [114]. Another resampling method is permutation testing, where labels are switched on data points when performing the test statistics. Both bootstrapping [32,78] and permutation tests [82,92] have been applied with HS-GC-IMS and NTS.

The success rate of different chemometric models for classification depends on many factors, such as botanical origin of the samples, number of samples or the selection of PCs, PLS, and k values. Gerhardt and coworkers compared different chemometric methods for the classification of the botanical origins of honey (acacia, canola, and honeydew) using a resolution-optimized HS-GC-IMS. They found a 98.6% accuracy with PCA-LDA, 86.1% with *k*NN (*k* = 5), and 97.0% with PLS-DA after employing 10-fold CV [52]. Quality assessment of olive oils based on temperature-ramped HS-GC-IMS and sensory evaluation conducted by Gerhardt and coworkers reported a 83.3% accuracy with PCA-LDA, 73.8% with *k*NN (*k* = 5) and 88.1% with SVM models, after employing 10-fold CV [79].

Artificial neural networks (ANNs) are a powerful modeling approach, which is vaguely inspired by the biological neural networks in the brain [115]. Due to their hidden layers, ANNs have the ability to capture complex interactions present in biological samples. Furthermore, ANNs can be applied for pattern recognition, classification tasks, and quantification problems, as well as data preprocessing. ANN has recently the gained attention for applications in food science and technology, while often being limited by the requirement of having sufficiently large data sets [116,117]. Zhu and coworkers have shown the superior prediction performance of ANN (89.5%) over PCA-LDA (65.7%), PLS-DA (58.7%), *k*NN (*k* = 5, 60.8%) and SVM (51.8%), and XGBoost (81.8%), for the classification of Sauvignon Blanc via SHS-GC-IMS [31]. Vega-Márquez and coworkers evaluated a deep learning network and five different benchmark methods for the classification of olive oil samples into EVOO, OO, and LVOO, based on HS-GC-IMS spectra obtained for 701 olive oil sample from two different harvests [91]. Among the five benchtop models used for comparison, XGBoost offers the best accuracy of 85.7%, compared to SVM (83.1%), *k*NN (84.5%), Tree (78.3%), and Regressor (85.5%). However, even better results were achieved with the deep learning approach, obtaining an accuracy of 88.8%, which underlines the potential of ANNs.

#### 3.2.3. Quantification Tasks

For quantification tasks, partial least squares regression (PLSR) has become the standard method used in chemometrics, including the fields of sensorial analysis in food chemistry [107,110]. PLSR is used to describe the relationship between two data matrices, X (experimental data) and Y (actual concentrations), which are decomposed into X = TP^T^ + E and Y = UQ^T^ + F, by finding the maximum covariance and linear relationship between the score matrices T and U. P and Q represent the loading matrices and E and F the matrices of residuals. After multiplying X with a nonlinear function, linear PLS can be applied as described above. After the successful implementation of multivariate models for the interpretation of the training data, the model performance is commonly tested using test data. The model quality can be determined by different figures of merit, such as the determination coefficient (R^2^) and the relative percentage error of prediction (RE), as well as root mean square error (RMSE), systematic error (Bias), or standard error of prediction (SEP).

One limitation to PLSR is the typically lower performance on nonlinear and heteroscedastic data, as is partially the case for IMS data. Several studies have analyzed nonlinear IMS data, as shown by the quantification of histamine in tuna stomach [7] or allergenic fragrance compounds, such as citral, in complex cosmetic products [38]. Nonlinear relationships between the matrices T and U can be described by kernel PLSR (also known as nonlinear PLSR), where the data are transformed into higher-dimensional spaces using the kernel trick [118].

An alternative to PLSR or kernel PLSR is provided by multivariate curve resolution alternating least squares (MCR-ALS). MCR decomposes the initial data matrix D with n sample spectra and m data points in the spectra to D = CS^T^ + E. The matrices C (n × l) and S (l × m) represent the concentration and spectra profiles of D for l components, while the matrix E (n × m) contains the residuals not explained by C and S [119]. MCR-ALS may deconvolute overlaying spectra from coelution and reconstitute the pure spectra for quantitation. A comparison of UR, PLSR, MCR-ALS, and Kernel-PLSR for the quantification of fragrances using GC-IMS analysis was provided by Brendel and coworkers. For both, a mixture of geraniol and citral as well as a mixture of the citral and cinnamal, Kernel-PLSR demonstrated the superior ability for the quantification of the nonlinear relationships of GC-IMS data since Kernel-PLSR was able to significantly reduce the RE of prediction and increase R^2^ of calibration.

### 3.3. Model Interpretation

Once a robust and reliable model is built, whose predictive abilities are sufficient for the chosen task, the model should be checked for plausibility. Therefore, the influencing variables, also referred to as markers, are extracted, identified, and subsequently used for interpreting and explaining the model. By using PCA for marker extraction, the loadings obtained by multivariate analysis are projected backwards into the original data space and are subsequently evaluated either manually or automatically. This allows one to identify the signals with the main influence for separation in the respective principal components with relation to the original data (Figure 3). Another example is the use of PLS-DA, where volcano plots and variable importance in projection (VIP) analysis have been performed to determine the final markers for each discrimination task [56]. Compounds with a VIP score > 1 are generally considered as suitable markers [120].

Web-based platforms, such as MetaboAnalyst (http://www.metaboanalyst.ca, accessed on 1 July 2021) [121] and XCMS Online (https://xcmsonline.scripps.edu/, accessed on 1 July 2021) [122], are designed to handle comprehensive untargeted metabolomic data. Compared to GC-MS data, which include m/z information, GC-IMS data are intrinsically limited to (normalized) drift times and retention times. While MetaboAnalyst has also been used for the processing of GC-IMS data, including the preprocessing (normalization and scaling) of data [82], VIP analysis [83], and entire chemometric analysis [123], the lack of m/z information limits the use of these metabolomic data processing tools for compound identification. However, the combination of retention indices and normalized drift times is considered as a reliable alternative; as in particular, drift times are highly reproducible. By comparing markers obtained from the GC-IMS data to databases, which are often provided by IMS manufacturers, the substances of interest can be identified [124,125]. Furthermore, a search of the literature and subsequent confirmation by reference substances can be used for substance identification [126,127,128]. Since coelution and matrix effects can influence GC-IMS data, a common technique is to spike a complex sample with the pure substance. To increase the number of substances identified and to further increase the model’s accuracy, complementary data, such as GC-MS [123], can be used as described in Section 3.4. This procedure is optional, as reliable models for classification and quantification tasks can be built solely from HS-GC-IMS data [44,80]. Typically, these MS detectors are unit-resolved and, as such, again limited in their selectivity in comparison to high-resolution systems, such as TOF or Orbitrap systems. However, a full spectral interpretation is not always necessary in NTS approaches. Some studies solely detect markers without subsequent identification of substances [43,80,84].

NTS is a powerful approach for complex classification and quantification tasks. However, next to model overfitting, the transferability of a model to new data poses a major challenge. A dramatic example is given by Contreras and coworkers who analyzed 701 olive oil samples from the years 2014–15 and from 2015–16 for the classification into EVOO, OO, and LVOO, using HS-GC-IMS with NTS [33]. A model built with olive oil samples from 2014–15 obtained a prediction percentage of 67.8%. A better prediction success was achieved for a model built with olive oil samples from 2014–15 and 2015–16, obtaining 79.4% accuracy. However, when applying the NTS model built with samples from the year 2014–15 to predict the years 2015–16, a prediction success of only 36.0% was achieved, revealing the low transferability of the model. This is however not an effect solely to be attributed to GC-IMS but rather is a general issue of profiling techniques so far and is mainly driven by typically high analytical variance together with a high biological variance of the samples.

To achieve good predictive abilities and transferability of a model to new data, large sample sets containing independent and diverse samples are necessary; however, studies are often limited by sample availability. To overcome this issue and to reveal influencing factors, a comprehensive evaluation of the model should be performed. The identification of influencing substances, for instance, may reveal the characteristic compounds or classes of substances in which the classes differ. A subsequent comprehensive evaluation can determine which factors may be caused by systematic errors and which factors allow for true differentiation between classes. High predictive accuracies for classification and quantification tasks can be achieved without substance identification, yet a comprehensive evaluation of the model helps to detect strategic errors, such as a narrow sample distribution, and thus should always be part of NTS approaches. Due to its robustness and comparably low prices, HS-GC-IMS is suitable for benchtop profiling. In combination with NTS approaches, HS-GC-IMS data can be used for the implementation of models, which are suitable for classification and quantification task in various fields.

### 3.4. Complementary Data (Optional)

Due to the complexity of biological matrices, the substance identification with GC-IMS data alone can be challenging, which is why stand-alone IMS are rarely used to investigate the sample composition. Complementary techniques, such as GC-MS [129] or ^1^H NMR [130], are often used to identify decisive marker substances [131].

Next to marker identification, complementary data can be combined with GC-IMS data to build a multimodal model. The process of integrating multiple data sources, which is commonly known as ‘data fusion’, has the potential to increase model accuracy and reliability, while reducing interferences and error rates [132]. The process of data fusion is categorized as low-level, mid-level (or intermediate-level), and high-level data fusion, depending on the fusion strategies used [133]. In low-level (or data-level) fusion approaches, data from all sources are preprocessed, concatenated into a common data matrix, and subsequently analyzed using classical multivariate methods, such as PCA or PLS (see Section 3.2). Mid-level data fusion approaches, which are also referred to as feature-level data fusion, are based on the extraction of relevant features from each data source separately [87]. Several latent variables, such as the score values from PCA or PLS, are selected for this feature-based data fusion, concatenated, and subsequently analyzed using classical multivariate methods. The third fusion approach is high-level or decision-level data fusion, where completely independent models are calculated from each data set; this level is the most challenging due to the determination of the ideal parameters for each separate multivariate model. Careful considerations of parameters, together with the implementation of a voting or scoring scheme, which prioritizes results from different data sources, can provide a combined model, which outperforms the individual models [134].

Schwolow and coworkers showed that the data fusion approach significantly increased both the predictive power and the robustness of the resulting classification model for the determination of geographical origin of olive oil [81]. The prediction accuracy obtained for Fourier-transform infrared (FT-IR) data alone was 67% and for GC-IMS 78%, while resulting a perfect score for a low-level data fusion approach using the complementary chemical information provided by GC-IMS and FT-IR analysis. The extra effort needed for data fusion approaches, however, is not always rewarded. Gu and coworkers classified fungal growth on peanut kernels into potentially aflatoxigenic fungi and non-aflatoxin producing fungal species using HS-GS-IMS and fluorescence measurements [87]. The predictive accuracy for HS-GC-IMS measurements alone was 93.3% with an OPLS-DA model, while low-level data fusion reduced the accuracy to 90%. The predictive accuracy of the OPLA-DA after mid-level data fusion using VIP > 1.3 further decreased to 86.7%. Only a mid-level data fusion using 10 PCs increased the model accuracy to 96.7%.

Another approach to increase the discrimination power of a classification model is the parallel analysis of IMS and MS, which has recently gained attention for classification and quantification tasks [90,106,127]. Although IMS and MS cannot be seen as fully complementary methods, in particular due to the identical volatile fraction being monitored by both techniques [90], it was shown that the soft ionization and drift-time-based ion separation on the one hand and a hard ionization and m/z-based separation on the other hand improved substance identification in the case of coelution in hops analysis [127]. While complementary data in general can increase the interpretability and accuracy of a model developed with HS-GC-IMS, they are optional for NTS via GC-IMS.

## 4. Comparison of NTS and Targeted Strategies

An alternative approach to NTS is targeted screening. While NTS of HS-GC-IMS data uses no pre-existing knowledge and the entire spectral fingerprint is subject to data analysis, for targeted analysis, specific markers are chosen prior to the data analysis. The markers used for targeted screening can be either handpicked or mathematically determined [33,135]. One-way analysis of variance (ANOVA), using for example a Tukey’s test, is often applied to identify volatile compounds which exhibit significant differences, commonly quantified at a 5% significance level (*p* ≤ 0.05) [59]. Various other methods, such as Gabor filters, local binary pattern, Haar, and histograms of oriented gradients (HOG), have been proposed for feature extraction [136,137]. Chen and coworkers applied MPCA and HOG for feature extraction and data reduction of MCC-IMS data, with subsequent canonical discriminant analysis for the generation of nonlinear boundaries, for the successful quantification of the adulteration degree of canola oil. A predictive accuracy of more than 95.2% was reported for a PLS model, which was obtained using a train–test split of 70–30 [99]. Using targeted approaches and applying PCA-*k*NN, the same authors reported a successful classification of rapeseed oils according to their quality (grade 1–4) and a successful determination of vegetable oil according to its botanical origin (sesame oil, rapeseed oil, and camellia oil). For the classification of the rapeseed oil quality, the colorized differences method was applied to CC-IMS data, resulting in 34 peaks of interest and a predictive accuracy of 100% [138]. Furthermore, Otsu’s method and colorized differences method was used for automatic peak detection, resulting in 88 peaks of interest and a predictive accuracy of 98.3% for the classification of vegetable oil using MCC-IMS data [139]. The advantage of preselecting markers with significant differences is the simultaneous reduction of noise in the data, which, however, includes the risk of overlooking valuable information.

NTS approaches (spectral fingerprinting) have also been directly compared to targeted approaches (extraction of specific markers) for the analysis of IMS data. Garrido-Delgado and coworkers compared targeted and NTS approaches for the classification of olive oil into EVOO, OO, and LVOO, using data obtained by MCC coupled to IMS [78]. A PCA-LDA model was used for data reduction and data clustering, followed by *k*NN (*k* = 3) for classification, obtaining a prediction percentage of 79% for the targeted strategy and 85% for NTS strategy. For the classification of olive oil harvested in 2014–15, Contreras and coworkers obtained a prediction percentage of 56.9% for the targeted strategy and 67.8% for the NTS strategy [33]. An improved prediction success was achieved for models built with olive oil samples from 2014–15 and 2015–16, obtaining 74.3% for the targeted strategy and 79.4% for the NTS, hence suggesting superior abilities of the NTS approach versus the selection of specific markers. By contrast, the authors also reported that a targeted model built with samples from the years 2014–15 (prediction success of 51.6%) was superior to the NTS approach (prediction success of 36.0%) when applied to the years 2015–16. Both models built with the data from the years 2014–15 show weak prediction abilities for the prediction of samples from the following year, revealing some fundamental challenges in data science: the predictive ability of a model is highly dependent on the number of samples as well as on the sample diversity. Both approaches include the risk of overfitting to a specific problem [33].

Arroyo-Manzanares and coworkers likewise obtained superior classification accuracy using HS-GC-IMS for a model based on a targeted marker selection (100%) compared to a model based on the whole spectral fingerprint (90%) for the distinction between dry-cured Iberian ham from pigs fattened on acorns and pasture or on feed [89]. However, the model based on marker selection was built using OPLS-DA, while *k*NN (*k* = 3) and PCA-LDA were used for the model based on spectral fingerprints; hence, the differences in the predictability of the models may result from the use of different mathematical tools and do not provide inferences about targeted and non-targeted approaches. Gu and coworkers by contrast obtained better classification results with the NTS versus a targeted approach for distinguishing between fungal infections of wheat kernels, as well as for the quantification of fungal colony counts [88]. In conclusion, NTS and targeted screening approaches are both effective tools for data analysis, with different challenges and application areas.

## 5. Conclusions

HS-GC-IMS in combination with an NTS approach provides an effective tool for classification and quantification tasks not at least due to low maintenance and easy handling of the instruments [9]. In combination with HS-GC and machine learning, IMS has been demonstrated to be an effective technique for various classification and quantification tasks, such as for the determination of food authenticity as well as for the detection of food adulteration and food fraud. Furthermore, HS-GC-IMS has been applied in the field of process control, for example for the early detection of microbial contaminants or in the field of food safety for the detection of pesticides. HS-GC-IMS is also found in the field of quality assurance for the detection of food freshness or the detection of off-flavors. Recently, HS-GC-IMS in combination with deep learning approaches has shown promising results for further improvement of model accuracies. HS-GC-IMS is suitable for benchtop profiling, due to its robustness and comparably low purchase and running costs. For benchtop analysis, MDV-based methods for HS-GC-IMS devices can be implemented in the laboratory and subsequently applied for food analysis in various fields. A common approach for further market acceptance is the combination of IMS instrumentation with HS-GC-MS, which alone has already been applied to a wide variety of applications.

## Figures and Tables

**Figure 1 molecules-26-05457-f001:**
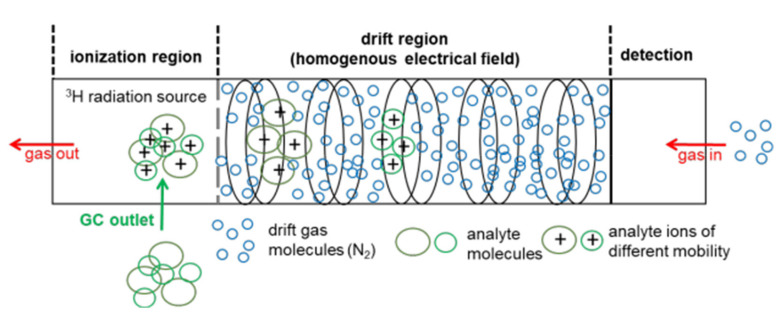
Setup of a drift-time IMS with a tritium (H-3) ionization source, adopted from [38] with permission (ID5138730886281).

**Figure 2 molecules-26-05457-f002:**
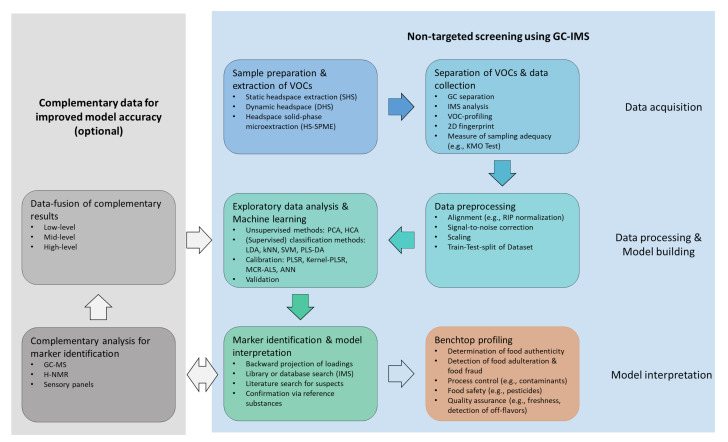
Workflow for non-targeted screening using HS-GC-IMS.

**Figure 3 molecules-26-05457-f003:**
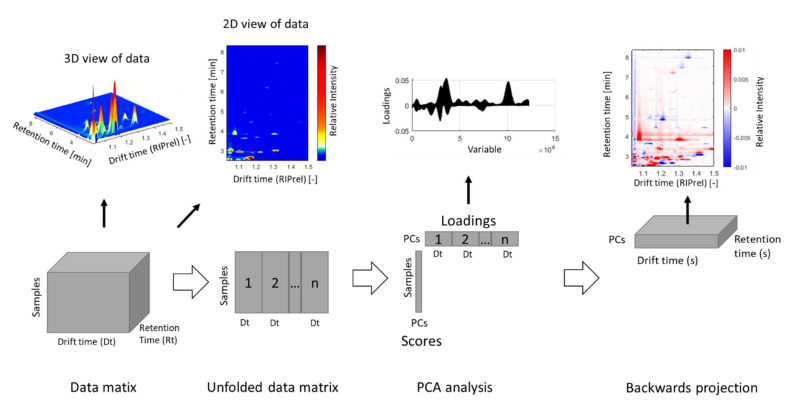
Data analysis and backwards projection of loadings for identification of key substances.

**Table 1 molecules-26-05457-t001:** Recent studies of HS-GC-IMS with NTS.

Reference (Main Author/Yeat)	Aim of Reference	Matrice (Number of Samples)	IMS Type (Ionization Source)	Separation (Length × Inner Diameter (ID), Film Thickness (ft))	Unsupervised and Supervised Methods	Complementary Analysis	Method and Number of Compounds Identified	Data-Split; (Cross-) Validation
Arroyo-Manzanares/2017 [89]	classification between pig’s food sources	dry-cured Iberian ham (24)	FlavourSpec by GAS (tritium, 6.5 keV)	FS-SE-54-CB stationary phase (30 m × 0.32 mm ID, 0.25 µm ft; 94% methyl, 5% phenyl, 1% vinyl silicone)	PCA PCA-LDA (90%), *k*NN (*k* = 3, 90%)		library search (LS) 5	80/20
Arroyo-Manzanares/2019 [84]	detection of adulteration (sugar cane or corn syrup)	honey (198)	FlavourSpec by GAS (tritium, 6.5 keV)	HP-5MS UI (nonpolar) (30 m × 0.25 mm ID, 0.25 μm ft)	OPLS-DA (97.4% prediction of class, 93.75% prediction of adulteration degree)		0	80/20
Brendel/2021 [90]	detection of botanical origin	citrus juice (47)	OEM IMS by GAS (tritium, 300 MBq)	ZB-5 ms column (30 m × 0.25 mm, 0.25 μm ft; 5% phenyl-methylpolysiloxane)	PCA PCA-LDA (91.5%)	HS-GC-MS/IMS	reference substances (RS) 9	leave-one-out (LOO) 4- and 6-fold CV
Cavanna/2018 [85]	detection of food freshness	egg (132)	FlavourSpec	FS-SE-54-CB-1 stationary phase (15 m × 0.53 mm ID, 1µm ft; 94% methyl, 5% phenyl, 1% vinyl silicone)	PCA OPLS-DA (97%)		RS, SPME-GC-MS 5	leave-one-out CV
Chen/2020 [30]	detection of geographical origin	Chinese yellow wine (122)	FlavourSpec by GAS (tritium, 6.5 keV)	nonpolar column (30 m, 95% methyl, 5% phenyl)	PCA QDA (95.35%)		LS 12	70/30
del Contreras/2019 [33]	quality assessment/classification	olive oil (701)	FlavourSpec by GAS (tritium, 6.5 keV)	SE-54-CB (30 m × 0.32 mm, 0.25 µm ft; 94% methyl, 5% phenyl, 1% vinyl silicone)	PCA non-targeted (PCA-LDA, *k*NN *k* = 3: 79.4%), targeted OPLS-DA: 74.3%)		LS, RS 20	80/20; 7-fold CV
Garrido-Delgado/2011 [80]	quality assessment/classification	olive oil (49)	portable UV–IMS instrument (UV ionization source: 10.6 eV) and FlavourSpec by GAS (tritium, 6.5 keV)	MCC (nonpolar) OV-5 (20 cm)	PCA UV-IMS: *k*NN (*k* = 3, 86.1%); GC-IMS: *k*NN (*k* = 3, 100%)		0	71/29 and 64/26; leave-one-out CV
Garrido-Delgado/2012 [78]	quality assessment/classification	olive oil (98)	FlavourSpec by GAS (tritium, 6.5 keV)	MCC (nonpolar) OV-5 (20 cm, 1000 parallel glass capillaries)	PCA Targeted: *k*NN (*k* = 3, 79%), non-targeted: *k*NN (*k* = 3, 87%)	Organoleptic analysis	LS, RS 10	bootstrap validation (B = 100)
Garrido-Delgado/2015 [32]	quality assessment/classification	olive oil (55)	FlavourSpec by GAS (tritium, 6.5 keV)	two different types of columns: MCC (20 cm × 3 mm ID, 900 parallel glass capillaries: 40µm ID, 0.2µm ft) and CC (30 m × 0.25 mm, 0.5µm ft)	PCA MCC-IMS: *k*NN (*k* = 3, 79%); CC-IMS: *k*NN (*k* = 3, 83%)		LS, RS 26	bootstrap validation
Gerhardt/2017 [34]	detection of geographical origin	olive oil (40)	FlavourSpec by GAS (tritium, 6.5 keV)	NB-225 (25 m × 0.32 mm × 0.25 μm ft; 25% phenyl, 25% cyanopropyl methyl siloxane)	PCA PCA-LDA (98%) and *k*NN (*k* = 5, 92%)		RS 4	10-fold CV
Gerhardt/2018 [52]	detection of botanical origin	honey (74)	advanced IMS by GAS (tritium, 300 MBq)	DB-225 (25 m × 0.32 mm, 0.25 μm ft; 25% phenyl, 25% cyanopropyl methyl siloxane)	PCA PCA−LDA (98.6%), *k*NN (*k* = 5, 86.1%), PLS-DA (PLS = 5, 97.0%)	^1^H NMR	RS 18	87/13; 10-fold CV
Gerhardt/2019 [79]	quality assessment/classification	olive oil (94)	FlavourSpec by GAS (tritium, 6.5 keV)	DB-225 (25 m × 0.32 mm × 0.25 μm ft; 25% phenyl, 25% cyanopropyl methyl siloxane)	PCA, HCA PCA-LDA (10 PCs, 83.3%), *k*NN (73.8%), SVM (88.1%)	Organoleptic analysis	RS 25	10-fold CV
Gu/2020 [88]	detection of quality changes during storage	wheat kernels infected with mold (90)	FlavourSpec by GAS (tritium, 6.5 keV)	MXT-WAX (nonpolar) (30 m × 0.53 mm ID, 0.1 μm ft)	PCA, HCA GA-SVM, Artificial samples: (Classification: 100%), (Quantification > 93.9%); Natural samples: (Classification: 60–86.7%, Quantification: 72.6–88.9%)		LS 13	67/33
Gu/2020 [86]	detection of quality changes during storage	milled rice infected with Aspergillus species (108)	FlavourSpec by GAS (tritium, 6.5 keV)	SE-54-CB	PCA *k*NN (*k* = 3, 94.44%), PLSR (90.9%)	electric nose (E-nose)	LS, partially RS 25	67/33; 10-fold CV
Gu/2021[87]	detection of quality changes during storage	peanut kernel infected with aflatoxigenic fungi (180)	FlavourSpec	Rtx-WAX (30 m × 0.32 mm ID, 0.25 μm ft, RT-12,424)	PCA OPLS-DA (93.3%), low-level data fusion (90%), mid-level data fusion (96.7%)	fluorescence analysis	LS 3	67/33; 7-fold CV, 200× permutation test
Li/2019 [5]	classification between aroma	green tea (23)	real-time IMS: self-made positive photoionization (PP-IMS) with KR lamp	-	PCA PLS-DA (95.6%)	sensory evaluation	0	10-fold CV, 1000× permutation test
Schwolow/2019 [81]	detection of botanical and geographical origin	honey and olive oil (64 honey, 54 EVOOs)	advanced IMS by GAS (tritium, 300 MBq)	DB-225 (25 m × 0.32 mm, 0.25 μm ft)	PCA Honey: PCA-LDA (33%); Olive oil: PCA-LDA (78%), PCA-LDA + data fusion (100%)	FT-MIR	previous studies (18 honey, 7 olive oil)	87/13; 10-fold CV
Shuai/2014 [44]	detection of adulteration (with vegetable oils)	flaxseed oil (78)	IMS-KS-100 by Wuhan Syscan Technology (pulse glow discharge)	n-hexane extraction	recursive support vector machine (R-SVM) (93.1%)		0	10-fold CV
Vega-Marquez/2019 [91]	quality assessment/classification	olive oil (701)	FlavourSpec by GAS (tritium, 6.5 keV)	SE-54-CB (30 m × 0.32 mm, 0.25 µm ft; 94% methyl, 5% phenyl, 1% vinyl silicone)	Deep learning (88.8%), SVM (83.1%), *k*NN (84.5%), Tree (78.3%), Regressor (85.5%), XGBoost (85.7%)		LS, RS 20	80/20
Wang/2019 [82]	detection of botanical origin	honey (40)	FlavourSpec by GAS (tritium, 6.5 keV)	FS–SE–54–CB-0.5 (15 m × 0.53 mm ID)	PCA OPLS-DA (95%) (VIP > 1.5)		HS-SPME-GC-MS8	200× permutation test
Wang/2019 [83]	detection of botanical origin	honey (120)	FlavourSpec by GAS (tritium, 6.5 keV)	FS–SE–54–CB-0.5 (15 m × 0.53 mm ID)	PCA PLS-DA (84%)		LS 25	CV-ANOVA
Yuan/2020 [92]	metabolomic studies	rat feces (30)	FlavourSpec	FS-SE-54-CB-1 (15 m × 0.53 mm ID)	PCA OPLS-DA (86.6)		LS 11	200× permutation test
Zhu/2020 [31]	quality assessment/classification	wine (143)	FlavourSpec	MXT-WAX (polar) (30 m × 0.53 mm ID, 0.5 μm ft, polyethylene glycol)	PCA PCA-LDA (65.7%), PLS-DA (58.7%), *k*NN (*k* = 5, 60.8%), SVM (51.8%), XGBoot (81.8%), ANN (89.5%)	sensory evaluation	RS >30	85/15; 10-fold CV

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
