# Peer review of "Non-Targeted Screening Approaches for Profiling of Volatile Organic Compounds Based on Gas Chromatography-Ion Mobility Spectroscopy (GC-IMS) and Machine Learning"

_molecules, 2021, doi:10.3390/molecules26185457_

Round 1

Reviewer 1 Report

This manuscript is a Review Article with the title “Non-targeted screening approaches for profiling of volatile organic compounds based on gas chromatography ion mobility  spectroscopy (GC-IMS) and machine learning”. The authors propose to explore the potential value of non-targeted screening (NTS) approaches on GC-IMS. It is well structured, the bibliography is complete and up-to-date.

I would simply recommend that authors include a glossary with all abbreviations and acronyms that appear in the text.

There are a few deficiencies in this article need to be improved according to following list of comments:

L. 22-156: In the section of INTRODUNCTION, the authors described in detail the principle of IMS. The schematic diagram of the principle of IMS can be added to make the reader better understand. In addition, the advantage and disadvantage of GC-IMS could be summarized as a table or picture.
L.199: In Table 1, the aim of the reference should be added.
L. 627-944: The format of the references needs to be modified. There was lack of volume and page numbers.

Author Response

Dear Editor and Referees, 
Thank you for your comments on our paper. The following outline addresses the changes that were made. For readability we have marked our comments in blue.  
Reviewer 1:  
This manuscript is a Review Article with the title “Non-targeted screening approaches for profiling of volatile organic compounds based on gas chromatography ion mobility spectroscopy (GC-IMS) and machine learning”. The authors propose to explore the potential value of non-targeted screening (NTS) approaches on GC-IMS. It is well structured, the bibliography is complete and up-to-date. I would simply recommend that authors include a glossary with all abbreviations and acronyms that appear in the text.  
A glossary with all abbreviations and acronyms was added before the Reference section. Furthermore, abbreviations were unified in the text. 
There are a few deficiencies in this article need to be improved according to following list of comments: 
➢ L. 22-156: In the section of INTRODUNCTION, the authors described in detail the principle of IMS. The schematic diagram of the principle of IMS can be added to make the reader better understand. In addition, the advantage and disadvantage of GC-IMS could be summarized as a table or picture. 
A schematic diagram of the principle was added in Figure 1 “Setup of a drift time IMS containing a tritium (H-3) ionization source.”. In addition, a paragraph highlighting important advantages and disadvantages of GC-IMS was added (L. 151-165).  
➢ L.199: In Table 1, the aim of the reference should be added. 
The aim of the reference was added in the second column. 
➢ L. 627-944: The format of the references needs to be modified. There was lack of volume and page numbers. 
To include volume and page numbers, the citation style was updated to ACS American Chemical Society, 4th ed.

Reviewer 2 Report

In this review, the authors revisited the state-of-the art of non-targeted screening approaches for profiling of volatile organic compounds based on GC-IMS. Important steps involved in the process such as  data acquisition, data processing and model building, model interpretation and complementary data analysis are extensively discussed. I strongly believe that this review will add a significant contribution to the field. Therefore, I recommend publication in its current form.

Author Response

Thank you for your feedback.